# Implementing an Early Detection Program for Autism Spectrum Disorders in the Polish Primary Healthcare Setting—Possible Obstacles and Experiences from Online ASD Screening

**DOI:** 10.3390/brainsci14040388

**Published:** 2024-04-16

**Authors:** Mateusz Sobieski, Urszula Grata-Borkowska, Maria Magdalena Bujnowska-Fedak

**Affiliations:** Department of Family Medicine, Wroclaw Medical University, Syrokomli 1, 51-141 Wroclaw, Poland; urszula.grata-borkowska@umw.edu.pl (U.G.-B.); maria.bujnowska-fedak@umw.edu.pl (M.M.B.-F.)

**Keywords:** autism spectrum disorders, population screening, autism screening, online screening

## Abstract

A screening questionnaire for autism symptoms is not yet available in Poland, and there are no recommendations regarding screening for developmental disorders in Polish primary healthcare. The aim of this study was to assess the opinions of parents and physicians on the legitimacy and necessity of screening for autism spectrum disorders, potential barriers to the implementation of the screening program, and the evaluation and presentation of the process of online ASD screening, which was part of the validation program for the Polish version of one of the screening tools. This study involved 418 parents whose children were screened online and 95 primary care physicians who expressed their opinions in prepared surveys. The results indicate that both parents and doctors perceive the need to screen children for ASD in the general population without a clear preference as to the screening method (online or in person). Moreover, online screening is considered by respondents as a satisfactory diagnostic method. Therefore, online screening may prove to be at least a partial method of solving numerous obstacles indicated by participants’ systemic difficulties including time constraints, the lack of experienced specialists in the field of developmental disorders and organizational difficulties of healthcare systems.

## 1. Introduction

Autism spectrum disorders (ASD) are neurodevelopmental conditions of undetermined etiology, most often manifested in early childhood, which affect daily activity and are characterized primarily by difficulties in the sphere of communication and interpersonal interactions as well as restricted interests or repetitive behaviors, with prevalence estimated to 1/100 worldwide [1,2]. There are no reliable data available on the prevalence of ASD in Poland—according to official data from the Polish National Health Fund from 2012, the prevalence rate of ASD in individuals under the age of 18 in Poland is 3.4 cases per 10,000 children [3]. More recent preliminary data from two Polish provinces estimate that ASD occurs in one in 286 children [4]. It is not entirely clear why the reported prevalence of ASD in Poland is lower than in neighboring countries (e.g., in Germany—0.38%) and lower than assumed, based on newer data from European Union countries [5,6]—it may be the result of lacking readily available tools for the screening and diagnosis of ASD, insufficient awareness of this problem of healthcare professionals (HCPs) and systemic difficulties and constraints.

Accelerated diagnosis of ASD enables initiation of an early, appropriately adjusted therapy [7]. First symptoms of ASD usually appear during infancy or early childhood—prodromal symptoms may be visible even as early as 6 months of a child’s life [7,8] and usually reliable diagnosis of ASD in a child can be made as early as 2–3 years of age [9]. The younger the child, the better the results of therapy that can be achieved in the area of communication and social interaction, cognitive abilities, speech development or behavior appropriate to the situation, which improves the quality of life of people with ASD, reduces the risk of mental disorders, and significantly reduces the ASD burden [10,11,12,13]. The burdens associated with the diagnosis of ASD in a child in the family include lowering the quality of life of the parents of a child with ASD, lowering the quality of interaction with the child and increasing perceived parental stress [14,15].

For this reason, the American Academy of Pediatrics recommends screening at 18 and 24 months of age as part of primary care during well-child care visits [16]. It appears that the increasing availability of screening significantly lowered the age of ASD diagnosis in the US, with diagnosis before the age of 4 made in 71% of children (2018) compared to 58% in 2014 [17,18]. However, a US Preventive Services Task Force from 2016 report shows insufficient evidence to recommend universal ASD screening [19]. On the other hand, there is evidence suggesting that including screening tools in routine medical appointments may result in earlier and more accurate identification of children who need further help than relying solely on clinical impressions, which is particularly important when care providers are less experienced in diagnosing ASD [20]. Moreover, the use of public ASD screening may reduce social inequalities in terms of the age of diagnosis and access to further therapeutic activities [21,22]. The conclusions of both reports indicate the need for further research on screening tools and their effectiveness, as well as on the effectiveness of further proceedings after screening [23].

Throughout infancy and early childhood, primary care professionals are the professionals with whom a young child most often comes into contact [24]. During this time, numerous meetings with doctors, nurses, midwives and other HCPs take place—as a result of visits due to the child’s illness, well-child check-ups or qualifying for vaccinations [25]. This enables close observation of development of the child and detection of developmental disorders [26]. Primary care professionals are believed to have the greatest influence on the early diagnosis of ASD (especially due to performing screening for developmental disorders (DD)), and they are responsible for the significant trend in the decline in the average age of children when diagnosed with autism spectrum disorders [26]. In addition, family doctors are coordinators of diagnostics and treatment performed by many of the further specialists, as well as “caregivers” of their patients not only on the health issues, but also on the social issues—which is necessary in the event of ASD in a child.

In Poland, well-child care visits have been conducted in primary healthcare clinics since the 1950s. The system of preventive healthcare for children is based on the principle of continuity of care—according to this assumption, the same doctor looks after a child in health and in sickness. After the newborn is discharged from the hospital, parents are obliged to report the child to a selected primary care clinic. Well-child care visits take place at the ages of 2, 6, 9, and 12 months and later at 2, 4 and 5 years of age. During the latter, an indicative examination of hearing (assessment of hearing behavior and the “show what you hear” test) and vision (especially in the direction of strabismus, later—also other disorders) is performed [27]. At 5–6 years of age, an indicative study of motor and psychosocial development should be carried out on the basis of interviews and observation, but it is rarely a validated, structured study [28]. Until then, the assessment of psychomotor development is primarily an assessment of the child’s time to reach milestones—there are no other established guidelines for the use of any tools for DD screening in children. In the case of suspicion of DDs, the child is referred to appropriate psychological and pedagogical clinics, where further examinations and consultations are carried out [29]. The validated diagnostic methods available in Poland include Level 2 tools, Autism Spectrum Rating Scales (ASRS) and Social Communication Questionnaire (SCQ), and Level 3 tools, Autism Diagnostic Observation Schedule (ADOS-2) and Psychoeducational Profile, Third Edition—PEP-3-PL [30,31,32,33]. Research on the Polish version of Autism Diagnostic Interview-Revised (ADI-R) is in progress [34]. During the final diagnosis, children with visible developmental disorders may be covered by the Early Development Support program—comprehensive, multidisciplinary activities to support the child’s psychomotor development through the intervention of therapists (psychologists, speech therapists, educators, SI therapists); however, they must be referred there by a doctor or preschool teacher.

In order to increase the effectiveness of early detection of autism spectrum symptoms, many tools have been created to facilitate this process for healthcare workers. Due to the lack of standardized tools for ASD screening in Poland, we started the “Spojrzeć w oczy” (eng. “Look in the eyes”) project. The aim of this project was to validate the Polish version of the Communication and Symbolic Behavior Scale-Developmental Profile Infant-Toddler Checklist (CSBS-DP ITC). The CSBS-DP ITC is one of the available tools for the early detection of symptoms of autism spectrum disorders consisting of 24 items, filled out by parents and guardians [35]. The CSBS-DP ITC can be used in ASD screening for the general population of children aged 6 to 24 months in a primary care setting. In previous studies, we showed that the Polish version of the CSBS-DP ITC has good psychometric properties and relatively high specificity and sensitivity and can be used as an effective screening tool [36,37].

During the planning of this study, the COVID-19 pandemic spread globally, which prevented efficiently carrying out the stationary study in primary healthcare clinics. The pandemic has made it necessary to introduce solutions that would allow access to remote medical services, which reduced the risk of coronavirus infection [38,39]. For this reason, it was necessary to create an electronic version of the CSBS-DP ITC questionnaire to be completed by parents at home. Then, the results of examination, instead of being assessed by individual healthcare professionals (HCPs, e.g., GPs, pediatricians, nurses, psychologists), were assessed directly by specialists involved in the project. Epidemiological situation forced the screening to be carried out in a unique way, using the online version, without direct contact with the examined person. The use of telemedicine made it possible to contact the patient, perform the screening, conduct follow-up tests and possibly dispel the parents’ doubts and questions. 

The aim of this paper is to assess opinions of HCPs and parents performing electronic screening and to present the experiences from one of the first attempts to conduct such a screening program. In addition, we wanted to assess the possible constraints in usage of ASD screening tools in HCPs’ everyday practice. Approval from the Bioethics Committee of the Wroclaw Medical University was obtained to conduct this study (number KB—641/2020). All procedures were performed in accordance with the 1964 Helsinki declaration and its later amendments.

## 2. Materials and Methods

### 2.1. Assessment of ASD Electronic Screening by Parents

We sent e-mail invitations to 1461 parents of children whose applications in the first phase of the “Spojrzec w oczy” project were included in the assessment of the psychometric values of the CSBS-DP ITC (the criteria for inclusion of children in this study regarding the properties of the questionnaire were being a resident of Poland, the age 6–24 months and usage of Polish as the main language by both parents) [36]. We asked parents to fill out an electronic short questionnaire about their feelings on electronic screening for developmental disorders, recording their age, level of education, place of residence and subjective information technology (IT) proficiency on a scale from 1 to 10 (where 1—is the lowest, 10—the highest) as factors potentially influencing the assessment of the diagnostic process [40,41]. A translated version of the survey is available as Appendix A. We did not link parents’ answers to their child’s results in the CSBS-DP ITC questionnaire to maintain anonymity; however, we asked them to select if they benefited from additional contact with people involved in the project (e.g., further diagnostics, specialist advice) or if their child was diagnosed with ASD or other developmental disorders, as a factor that could potentially increase satisfaction with online screening. The survey was made available on the project’s website—it is available only through the appropriate link, sent only to the parents included in this study. Invitations to evaluate the project were sent on 9 January 2023 and responses were collected until 18 January 2023. 

### 2.2. Assessment of ASD Electronic Screening by Doctors and Potential Difficulties in Implementing the ASD Screening Program

In order to recruit physicians for this study, we sent e-mail invitations to members in the Polish Society of Family Medicine and we propagated the survey on the project’s social media profiles. The doctors were verified with their profession practice number—a document possessed by every doctor practicing in Poland. The survey consisted of 18 questions, 7 of which concerned statistical data—age, sex, main place of work, work experience, and the percentage of children with developmental disorders (DD) under their own medical care. The remaining ones touched upon aspects of the use of diagnostic methods in clinical practice, further proceedings when DD is suspected, the willingness to use screening methods during medical visits, the assessment of potential difficulties and limitations in introducing screening tests, and the positives and negatives resulting from online DD screening. A translated version of the survey for doctors is available as Appendix A. Applications were collected from 31 July 2023 to 1 February 2024.

### 2.3. Statistical Analysis

All the analyses presented in this manuscript were performed using Statistica 13.3 software. The Shapiro–Wilk normality test was applied to check the normal distribution using 0.05 as a significance level. None of the examined variables met the criterion of normality of distribution; therefore, non-parametric tests were used for further analysis. The dependence of the variables on the categories of qualitative variables was tested using the non-parametric Kruskal–Wallis test. This test enabled a direct comparison of the value of a quantitative variable between the two categories of a qualitative variable. In the case of qualitative variables, the uniformity of the category distribution was tested with the chi-square uniformity test (one sample proportions test). In some cases, Spearman’s rank correlation was used, allowing variables on ordinal and quantitative scales that did not have a normal distribution to be correlated with each other. In the case of a small number of respondents in subgroups, the Yates correction was used in the chi-square test. A significance level of 0.05 was set for all tests. In the case of some of the examined features, only a descriptive interpretation was possible. This is mainly due to the inability to conduct further statistical analyses—these variables cannot be tested for homogeneity. The respondents could indicate several categories at the same time, so the categories do not meet the separability condition required for homogeneity testing.

## 3. Results

We received a positive response and a fully completed survey from 418 parents taking part in the “Spojrzeć w oczy” project, giving a response rate of 28.6%. Due to forcing the answers to individual questions and the full anonymity of the survey, it was not necessary to exclude any submissions. The number of participants exceeds the minimum sample size to achieve a confidence level of 95% and a margin of error of 5% in the Polish population (the calculated minimum sample size was 384) [42]. All submissions were completed by female participants except one. The mean age of the parents was 33.86 years (SD = 4.12) and the mean subjective IT proficiency was 8.52 (SD = 1.25). Parents’ full sociodemographic characteristics are presented in Table 1. 

Among the parents who took part in the ASD electronic screening assessment, 38 additionally contacted us for more information and diagnostic and therapeutic assistance via e-mail and telephone. A total of 30 of their children were diagnosed with a developmental disorder—21 with ASD, 8 with speech development delay, and 1 with sensory integration disorder. 

Of all the parents, only 4 (0.95%) had not heard of ASD before and 184 (44.02%) did not suspect any developmental disorders in their child—in this group, one child was diagnosed with ASD, the other 3—with language delay (LD). Each parent was asked to answer questions about ASD awareness and opinions on ASD screening—both online and in person. Chi-square tests and the Kruskal–Wallis test were used for this analysis, depending on the nature of the studied variables. The collected results are presented in Table 2 and Table 3.

The significant majority of parents (99.04% of all, N = 414) participating in this study were aware of the issue of ASD at least to a basic extent before the screening test. Nevertheless, there is a significant difference in the level of awareness of ASD among parents with lower education, living in rural areas and with lower IT proficiency compared to other parents (χ^2^ = 13.013, *p* = 0.005; χ^2^ = 13.398, *p* = <0.001; H = 14.801, *p* = 0.011, respectively). Moreover, as many as 98.09% of parents (N = 410) believe that ASD screening should be mandatory, 98.56% (N = 412) would participate in screening for developmental disorders in their next child again, and 97.13% (N = 406) would participate in online screening. The percentage of parents willing to participate again for another child is lower for parents living in smaller towns. The parents’ opinion differs widely in the case of preferences for the form of screening (online or stationary)—a slight majority prefer the online version (52.15%, N = 218), but this does not depend on place of residence, education or the fact of receiving a diagnosis of developmental disorders in a child—the only factor influencing the preference for the online method is higher IT proficiency (H = 16.212; *p* = 0.006).

A rather intriguing issue is the fact that parents whose children were finally diagnosed with developmental disorders report much more often than others that they have had the thought that their child may be at risk of suffering from ASD (χ^2^ = 12.350, *p* = <0.001).

We also asked parents four questions regarding their assessment of the electronic screening carried out by us as part of the “Spojrzeć w oczy” project using the Polish online-version of the CSBS-DP ITC questionnaire. Responses to questions were based on a five-point Likert scale. The Kruskal–Wallis test and Spearman’s rank correlation were used for this analysis. The results regarding the assessment of the quality of online screening and satisfaction with participation in this study depending on age, IT proficiency, place of residence and education are included in Table 4 and Table 5.

Collected data indicate a very high overall assessment of the online screening conducted as part of the project by parents. Opinions are characterized by high homogeneity in terms of the examined features, with a few exceptions. Parents living in small towns rated the possibility of contact and the possibility of receiving help the lowest (H = 10.794, *p* = 0.013 and H = 21.323, *p* = <0.001, respectively); similarly, people with higher education rated the possibility of obtaining appropriate help lower than parents with secondary education (H = 6.043, *p* = 0.048). Moreover, there is a very weak negative correlation indicating that older parents evaluate the possibility of obtaining further information and answers from examiners during online screening lower than younger parents (R = −0.104, *p* = 0.033).

In order to obtain doctors’ opinions on electronic ASD screening and potential limitations in the implementation of ASD screening in everyday practice, doctors from the Polish Society of Family Medicine were invited to participate in the second part of this study. Finally, 95 doctors took part in this study. The exact number of Society members is not officially available; according to Facebook data, the post reached approximately 1320 users. Mean age of the doctors was 32.58 years (SD = 5.24) and mean subjective IT proficiency was 8.33 (SD = 1.32). The average length of service in primary healthcare facilities of the respondents was 5.23 years (SD = 4.68). Full doctors’ sociodemographic characteristics are presented in Table 6. 

Physicians were asked about their own methods of management of a pediatric patient in case of suspected developmental disorders—whether they rely on their own clinical assessment or the results of additional tests when choosing further treatment, and whether they use screening methods in their everyday work. In addition, we asked for information on whether they use given methods in all children or only those from the risk group, and what actions are taken when observing DD in a patient; finally, we asked doctors to choose their preferred ASD screening option. Chi-square tests and the Kruskal–Wallis test were used for this analysis, depending on the nature of the studied variables. The collected data are presented in Appendix A, Table 7 and Table 8. Appendix A is provided as Appendix A.

The vast majority of surveyed physicians use basic methods to detect symptoms of developmental disorders—the most common ones include observing the child during the physical examination (94.74%), attempting to communicate and establish contact with the child (93.68%), and assessing the pace of achieving developmental milestones (89.47%). Risks related to family history are slightly less frequently taken into account; diagnostic tools completed by caregivers or healthcare workers are used much less frequently (21.05 and 15.79% of respondents, respectively). Moreover, active use of the above-mentioned methods is more common only in children suspected of having DD features; less frequently in the entire pediatric patient population. Quite a significant part of the surveyed doctors has a wait-and-see attitude in the case of subtle features of DD in a child under two years of age (37.89%); in the case of children over two years of age, the vast majority of doctors refer children with DD symptoms for further evaluation (94.74%). These issues are independent of doctors’ age, gender, place of work and age; the analyzes showed the significance of the percentage of pediatric patients among all patients on the frequency of screening use in the entire population (the highest in the “up to 20%” and “above 50%” groups, not showing a linear nature; X^2^ = 19.216, *p* = 0.002), self-estimated IT proficiency on the frequency of referring patients for further evaluation after the age of 2) (the higher it is, the greater the percentage of referring physicians) and the length of work experience in primary care facilities (the longer it is, the more physicians refer patients for further evaluation—both below and over 2 years of age).

As in the case of parents, preferences regarding the type of screening (online vs. stationary) are strongly divided (53.68% of doctors are in favor of online screening). Dependencies were observed only in terms of age and work experience of doctors (younger doctors and those working in primary care for a shorter period more often preferred the online version (H = 27.876, *p* = 0.046 and H = 24.384, *p* = 0.041, respectively).

If the stationary version was preferred over the electronic version, we asked doctors who would be responsible for calculating the questionnaire results and analyzing the screening results. The vast majority (n = 36, 81.82%) believe that this should be performed by the child’s doctor; then the doctor with the most experience in DD (n = 6, 13.64%). Interestingly, none of the doctors would hand over this function to nurses or the clinic coordinator (in Poland, these are HCPs responsible for helping patients set appointments with specialists outside the clinic and helping with the care of chronically ill patients).

Physicians were also asked how willing they would be to use questionnaires to perform screening of DDs during individual visits (e.g., peri-vaccination visits, well-child visits, when observing DD symptoms during the visit). The Kruskal–Wallis test was used to perform this analysis. The results are summarized in Table 9.

The respondents’ answers show a clear, statistically significant difference in the willingness to use screening questionnaires during various situations in the physicians’ office (H = 46.069, *p* < 0.001); doctors are significantly more willing to use screening tools to confirm noticed symptoms (M = 4.362) and to confirm or dispel parents’ doubts (M = 4.362). The willingness to use these methods for general screening is significantly lower (in the case of 18 months—M = 3.512; 24 months—M = 3.828).

While preparing the Polish version of the CSBS-DP ITC questionnaire, we encountered many difficulties reported by healthcare professionals (HCPs) regarding the implementation of additional ASD screening. We collected their and our observations in order to obtain a broader opinion of HCPs potentially responsible for screening. The answers are summarized in Figure 1 and Appendix A. Appendix A is provided as Appendix A.

The surveyed physicians most often point out that there is insufficient time during the patient’s visit to be able to carry out screening test for developmental disorders—this answer was given by almost 92% of doctors. It is also clearly the most frequently occurring most serious obstacle according to doctors; almost half of the respondents point out that excessive time burden is the main barrier to conducting population screening tests. Slightly fewer physicians report difficulties in access to specialists (both the lack of specialists and queues to practicing HCPs) and the lack of clear clinical recommendations regarding DD screening.

Parents and doctors were also asked to rate whether they agreed with the opinions about the positives and negatives of online ASD screening. These opinions were collected based on contacts with parents, specialists and the authors’ own experiences. Chi-square test analysis was performed to obtain results. The respondents’ answers are presented in Table 10.

Both in the groups of doctors and parents, positive features of online screening were more often marked than negative ones. What parents most often appreciate are the convenience of online screening (saving time and the possibility of conducting the test from home—68.18% and 78.23% of parents, respectively) and easier access to the evaluation. Among the negatives, parents most often point out the inability to verify the result and assess the child’s development directly by a specialist (57.89%) and the lack of personal contact with examiners (46.65%). Similar results were obtained among doctors—they also noted the comfort of online screening; however, this effect was significantly lower than in parents (*p* = 0.025 and 0.002). Doctors also mainly pointed out the lack of direct contact with the person conducting the examination as main flaw of online screening (55.79%). What is noteworthy is that both groups (especially parents) rarely considered a potential lack of trust in people responsible for screening as a disadvantage.

## 4. Discussion

Conducted study provides evidence for potential benefits of using online ASD screening. Results indicate high parental satisfaction with participation in online screening, with high willingness to participate again for subsequent children. Moreover, most parents and doctors note the need to screen children for ASD. In both groups, there is no clear preference for the screening method (online vs. stationary). However, in order to implement any of the above-mentioned screening methods, the availability of linguistically and culturally adapted tools must be ensured.

In order to increase the effectiveness of early detection of DD, numerous screening questionnaires have been created, which, as research conducted in the United States shows, may be a useful tool to increase the percentage of early detected cases of developmental disorder [17]. In order to enable the use of this method to accelerate the diagnosis of developmental disorders, a version of the CSBS-DP ITC adapted to Polish conditions was prepared by authors of this paper, which was linguistically and culturally adapted to increase the effectiveness of the tool and reduce the likelihood of obtaining incorrect results. When designing the whole project methodology, it was necessary to adapt to the dynamically changing conditions during the COVID-19 pandemic. The pandemic forced the entire study to be conducted online or via telephone consultation. Nevertheless, this form of conducting this study made it possible—apart from assessing the effectiveness of the Polish version of the CSBS-DP ITC—to examine how parents evaluate on-line screening method. 

Preparing a Polish version of a fully validated ASD screening questionnaire could, at least partially, fill the gap related to the low frequency of use of diagnostic methods in the field of developmental disorders in the daily work of doctors, as evidenced by our study results. Polish doctors more often use their own clinical assessment of a child’s development, based on assessing the time of reaching appropriate milestones or observing the child’s behavior during visit, which, unfortunately, may result in a delay in diagnosis [17]. The use of tools in the practice of Polish primary care physicians is clearly lower than, for example, in the USA (where ASD screening programs are the most popular—63% of pediatricians use screening questionnaires in their practice, and up to 73% of all children are screened in this direction) which allows for lowering the average age of ASD diagnosis [43,44]. Another example of unintentional omissions may be the quite large percentage of Polish physicians observing the child’s development in the event of subtle symptoms of developmental disorders in children under two years of age—almost half of the respondents would continue close observation or refer the child for further evaluation based on parental pressure, which may also lead to a potential delay in the detection of developmental disorders. In the case of an older child (over 2 years of age), only 5% of respondents would decide to further observe the development. 

The most frequently raised issue by respondents, which may potentially hinder the implementation of a screening program for developmental disorders in Poland, is the insufficient amount of time that can be allocated for screening during a visit of a young patient. Paying attention to the occurrence of ASD symptoms in a child or conducting an appropriate screening test requires spending more time on these activities than is usually allocated to a visit to a primary care facility [45]. Insufficient time during the visit to conduct observations for ASD is also the most frequently mentioned obstacle in studies conducted in other countries, e.g., the USA, Canada or Oman [46,47,48,49]. The problem of time constraints is becoming more crucial—data from the UK show increasing workload for doctors and nurses working in primary care [50]. In Poland, this problem is probably even more severe due to the shortage of doctors and the resulting overwork—the number of practicing doctors and nurses in Poland is among the lowest in the EU and amounts to 2.4 doctors and 5.1 nurses per 1000 inhabitants; this problem is particularly severe in small counties around large cities and in rural areas [51]. However, there is a lack of precise data on the professional burden of Polish HCPs—in one of the studies conducted during the COVID-19 pandemic, Polish healthcare workers reported high levels of burnout and stress—related to, among others, increased workload [52]. Rising costs may also be a problem, including costs related to the preparation of materials for screening (although, as the collected data indicate, according to Polish doctors, this should not be a major obstacle) or, above all, with further care for children with suspected or ultimately diagnosed ASD. Healthcare financing in Poland is lower than the EU average (6.5% to 9.9% of GDP in 2019), most of which is spent on inpatient care; funds for outpatient care are half of the average in the European Union [51]. Low expenditure on outpatient healthcare and a small number of practicing HCPs are also related to the issue of difficulties in access to specialists in the immediate area, which is pointed out by as many as 54% of doctors—psychologists, speech therapists, educators, and child psychiatrists. Over the last decade (2014–2023), the number of child psychiatrists in Poland increased from 346 to 532, while in the years 2019–2023 there was an increase in the number of patients from nearly 150,000 to over 266,000, which means that the difficulties in obtaining appropriate specialist help are constantly increasing [53]. This is an important problem because experience from other countries (e.g., Taiwan) indicates that only increasing access to ASD screening without improving access to further evaluation or therapy may cause increasing frustration and confusion in families due to the lack of a coherent procedural and diagnostic system for people at risk of DD. To address the unmet needs of families with children with ASD, resource imbalances between screening and follow-up interventions in public pediatric care settings must be simultaneously addressed [54]. 

Another problem potentially troubling the implementation of screening is the lack of appropriate education of HCPs and the lack of systemic activities regarding the early diagnosis of ASD—gaps in knowledge regarding ASD and the ability to use screening tools are indicated by 33 and 36% of respondents, respectively. An even greater percentage of doctors report uncertainty regarding further treatment of a patient suspected of having ASD. Insufficient knowledge about ASD among physicians is a common global problem—a study conducted in 2020 showed that only 23% of primary care physicians had sufficient knowledge about ASD, and the percentage of such doctors was higher in countries with higher income [55]. This is probably due to the lack of experience in working with people with disabilities during medical studies, the small number of classes devoted to developmental disorders, as well as the specific image of people with ASD created by the mass media [56]. The problem of a lack of knowledge is intensified by the lack of clear guidelines (either Polish or European) regarding ASD screening and further diagnostic activities. Existing American guidelines do not fully correspond to the Polish primary care setting and their recommendations are difficult to implement into practice.

In Poland, fully validated screening questionnaires for ASD are virtually unavailable, which is indicated by 30% of respondents as a barrier to the implementation of screening. In addition to the mentioned work on the CSBS-DP ITC, so far, there are only preliminary data on the Polish version of Quantitative Checklist for Autism in Toddlers (Q-CHAT). Research on most frequently used in Poland M-CHAT organized as part of the Badabada project is still in progress [57,58]. To datew, direct translations of the above-mentioned questionnaires available on the Internet were usually used without any cultural and linguistic adaptation and without a validation process. Using incorrectly prepared diagnostic tools may reduce the accuracy of the diagnostic process; the diagnostic tool should be fully adapted to the population in which it will be used so that its psychometric properties are at the highest possible level [59]. Nevertheless, the vast majority of physicians would be willing to use diagnostic materials if they were available, especially when detecting clinically significant signs of developmental disorders or to allay or confirm parental concerns. Unfortunately, there is significantly less willingness in physicians to use these tools during peri-vaccination or well-child visits at 18 or 24 months of age, which would be the recommended course of action according to American Academy of Pediatrics guidelines [60]. Experience from preliminary screening programs in Spain and the Netherlands indicates that the use of screening methods for ASD during well-baby check-up visits increases the attendance rate [61]. Less willingness to conduct population screening may lead to delays in diagnosis, especially in the case of more subtle symptoms occurring in a child with undiagnosed DD.

Additionally, blurred responsibility for a child with DD makes it difficult to further guide the child in the diagnostic and therapeutic process. Care for a child with DD in Poland is divided into healthcare (within primary care clinics, specialist mental health clinics, community psychological care centers and early intervention centers) and educational care (within special/integrated kindergartens, psychological and pedagogical counseling centers, early childhood supporting child development and leading centers coordinating rehabilitation and care (pol. Wiodący Ośrodek Koordynujący Rehabilitacyjno-Opiekuńczy, WOKRO)). The multitude of facilities whose scope of competences are sometimes unclear and unregulated (e.g., early intervention centers do not have their own legal normative acts) means that obtaining appropriate, full assistance can be troublesome for child’s guardians who are placed in a stressful situation. The difficult situation is complemented by the fact that some of the mentioned forms of support are insufficient for all those in need—for example, in Poland, there are only forty early intervention centers [62]. The situation is further complicated by the fact that due to the underfunding of the public system, parents often use private healthcare. This significantly complicates the coordination of ASD diagnosis and therapy in Poland. Similar difficulties are also observed in other European countries—a survey conducted in south-eastern European countries indicates that many parents experienced difficulties or delays in therapy due to queues to specialists, high costs or difficulties in obtaining information; difficulties also concerned problems with obtaining assistance in the field of education [63]. 

The results of our study prove that respondents more often pay attention to the positive rather than negative effects of electronic screening. Both the groups of doctors and parents indicated primarily the convenience of this solution method, which is consistent with previous research results on the use of teleconsultations and online services in healthcare, which unanimously note greater availability and convenience of use compared to stationary services [64]. However, parents most often pointed out as a flaw that screening is carried out solely on the basis of a questionnaire completed by the parent, without direct contact with the examiner who could observe the child (and potentially confirm or deny the diagnosis resulting from screening). This is probably related to the subjective assessment of the doctor’s role as more important during the diagnostic process—not only related to the doctor’s experience, but also to interpersonal and communication qualifications, as well as empathy and understanding. Research data indicate that over 50% of patients do not fully believe in medical suggestions using artificial intelligence (AI), even though AI achieves better diagnostic results in some issues [65]. Probably, even in the case of a simple screening questionnaire (such as the CSBS-DP ITC), the issue of trust in the doctor is an important aspect for parents; it was proven that patients were more willing to implement recommendations given by doctors than those given by computers [66,67]. This effect may be even more important due to the fact that the diagnosis of ASD in a child is associated with a huge caregiving burden, even greater than for parents of children with Down syndrome or type 1 diabetes [68]. Preliminary information collected during this study indicates the need to take this fact into account when designing subsequent ASD screening services.

The collected data show that for parents the ability to initially confirm or deny fears is very important—despite this, the vast majority of parents believe that ASD screening should be mandatory for every child and that they would subject their next child to a similar test—regardless of the form of screening. Interestingly, both doctors and parents show an almost equal division in preferences for online or stationary screening. In the case of implementing the former in a larger population, attention should be paid to the possible effect of digital exclusion, observed more often among people with lower IT proficiency and from rural areas. Nevertheless, conducting diagnostic methods using electronic technologies potentially should speed up the provision of medical assistance and facilitate access to screening [69]. 

Despite all the positives regarding online screening described above, conducting screening with usage of online technologies is not yet popular—as of 29 November 2023, the phrase “electronic screening” is found in 275 and “online screening” in 258 scientific articles contained in the PubMed database. In the same database, only 2 studies on ASD screening can be found—one conducted in Italy, the other in China—assessing only the effectiveness of ASD online screening [70,71]. Additionally, one large study evaluates the effectiveness of screening using electronic techniques during inpatient visits [72]. We found no study addresses the issue of online screening assessment by parents or doctors. Our work is likely to be the beginning of a future discussion on the role of further online electronic services in the daily work of physicians.

The last issue worth mentioning is the fact that parents whose children were finally diagnosed with DD had previously suspected the occurrence of some developmental disorders in their children. Due to the fact that similar experiences also result from previous studies, it is worth paying attention in everyday practice to the possibilities of dispelling these concerns of caregivers as safely as possible—even in the case of negative tests and screening observations [73].

### Limitations

This study has several limitations. Firstly, parents’ assessment of satisfaction with the screening was in some cases carried out a considerable time after the end of participation in this study—this may result in an incomplete and slightly inadequate assessment. Moreover, the selection of the study group does not fully reflect the structure of Polish society. The answers in the section regarding parents’ assessment were provided almost exclusively by mothers—this is consistent with the sex structure obtained in previous studies on the diagnostic accuracy of the CSBS-DP ITC and is probably related to the culturally preferred family model in Poland, where mothers take most of the care of offspring, especially when children they are younger [74]. Percentage of people with higher education and living in large cities took part more often in the ASD screening during the “Spojrzeć w oczy” project; their participation increased even further in this study [75]. Similarly, the largest group of surveyed doctors are those living in large urban centers, with potential underrepresentation of rural areas of Poland. A large part of the surveyed physicians were younger doctors—residents—who are also overrepresented in the study group. Ultimately, both parents and doctors participated in this study completely voluntarily—there is a risk of biasing the group among people with a keen interest in the subject of ASD, which may result in inflated results regarding, for example, awareness of ASD issues or more frequent use of additional diagnostic materials. The methods used in the statistical analysis also pose certain limitations. Due to the lack of normal distribution of the studied features, it was necessary to use non-parametric tests with lower power than parametric tests. Moreover, due to the lack of homogeneity in the diagnostic methods used by doctors or potential restrictions on the implementation of ASD screening in the Polish primary care setting only a descriptive interpretation was possible to perform.

## 5. Conclusions

The data collected from our study clearly prove that Polish parents and doctors expect the implementation of screening for developmental disorders as part of routine child healthcare. However, this requires resolving numerous possible obstacles—primarily time constraints, human resources and systemic difficulties. Online screening could partially facilitate access to the test and reduce the workload, but it must be introduced with taking into account limitations—especially the possibility of quick verification of the screening result using more accurate methods during stationary visits. Moreover, it will not fully replace the doctor–patient relationship. However, it may be one of the ways to speed up the diagnostic and therapeutic process, which may improve the functioning of people affected by ASD.

## Figures and Tables

**Figure 1 brainsci-14-00388-f001:**
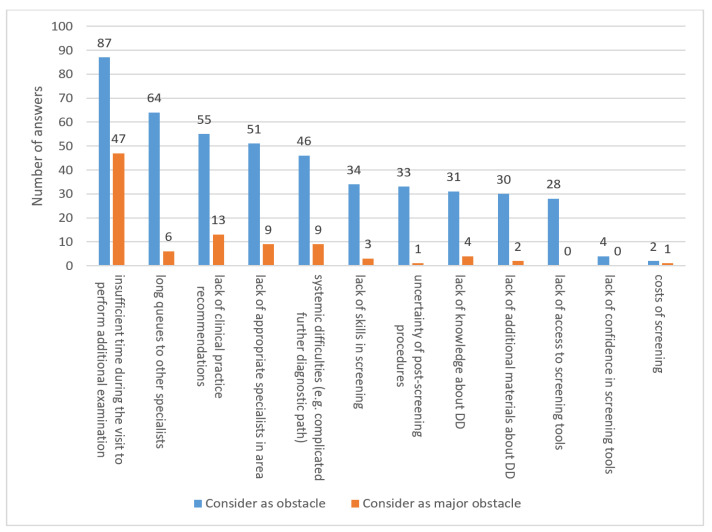
Obstacles to the implementation of screening in primary healthcare facilities according to surveyed physicians.

**Table 1 brainsci-14-00388-t001:** Sociodemographic characteristics of parents taking part in this study.

Characteristic	*n*	*%*
**Sex**		
Female	417	99.76%
Male	1	0.24%
Using additional contact with researchers		
Yes	38	9.09%
No	380	90.91%
Diagnosed DD in their child during the project		
Yes	30 (21 ASD, 8 LD, 1 SI)	7.18%
No	388	92.82%
Education level		
Lower education	0	0%
Secondary education	58	13.88%
Higher education	360	86.12%
Place of residence		
Village	99	23.68%
Town inhabited by less than 20,000 people	33	7.89%
City inhabited by 20,000–100,000 people	59	14.11%
City inhabited by more than 100,000 people	227	54.31%

**Note.** Secondary education is basic vocational, general or technical secondary education with or without a high school diploma and post-secondary studies which are not higher education). DD—developmental disorders.

**Table 2 brainsci-14-00388-t002:** Opinions on ASD screening (also online) and awareness of the problem among parents depending on place of residence, education and final diagnosis of the child.

	Number of Positive Responses	Number of Negative Responses	% of Positive Responses	*df*	Χ^2^	*p*
**Before screening, had you heard about ASD?**						
Place of residency					13.013	0.005
Village	95	4	95.96%	3		
Town inhabited by less than 20,000 people	33	0	100%			
City inhabited by 20,000–100,000 people	59	0	100%			
City inhabited by more than 100,000 people	227	0	100%			
Education level					13.398	<0.001
Secondary education	54	4	93.10%	1		
Higher education	360	0	100%			
Received diagnosis of DD						
Yes	30	0	100%	1	0.312	0.576
No	384	4	98.97%			
**Has it occurred to you that there may be possibility that your children may have ASD?**						
Place of residency					9.581	0.022
Village	50	49	50.51%	3		
Town inhabited by less than 20,000 people	23	10	69.70%			
City inhabited by 20,000–100,000 people	25	34	42.37%			
City inhabited by more than 100,000 people	136	91	59.91%			
Education level					0.189	0.663
Secondary education	32	26	55.17%	1		
Higher education	202	158	56.11%			
Received diagnosis of DD						
Yes	26	4	86.67%	1	12.350	<0.001
No	208	180	53.61%			
**Should screening for ASD be mandatory?**						
Place of residency					6.749	0.081
Village	99	0	100%	3		
Town inhabited by less than 20,000 people	31	2	93.94%			
City inhabited by 20,000–100,000 people	59	0	100%			
City inhabited by more than 100,000 people	221	6	97.36%			
Education level					1.028	0.311
Secondary education	56	2	96.55%	1		
Higher education	354	6	98.33%			
Received diagnosis of DD						
Yes	30	0	100%	1	0.631	0.427
No	380	8	97.94%			
**Would you participate in screening for ASD in your other children?**						
Place of residency					8.497	0.037
Village	99	0	100%	3		
Town inhabited by less than 20,000 people	31	2	93.94%			
City inhabited by 20,000–100,000 people	57	2	96.61%			
City inhabited by more than 100,000 people	225	2	99.12%			
Education level					0.913	0.339
Secondary education	58	0	100%	1		
Higher education	354	6	98.33%			
Received diagnosis of DD						
Yes	30	0	100%	1	0.471	0.493
No	382	6	98.45%			
**Would you participate in online screening for ASD in your other children?**						
Place of residency					11.429	0.010
Village	96	3	96.97%	3		
Town inhabited by less than 20,000 people	29	4	87.88%			
City inhabited by 20,000–100,000 people	58	1	98.31%			
City inhabited by more than 100,000 people	223	4	98.24%			
Education level					0.143	0.705
Secondary education	56	2	96.55%	1		
Higher education	350	10	97.22%			
Received diagnosis of DD						
Yes	28	2	93.33%	1	1.670	0.196
No	378	10	97.42%			
**Do you prefer screening to be carried out stationary at your clinic?**						
Place of residency					4.084	0.254
Village	42	57	42.42%	3		
Town inhabited by less than 20,000 people	19	14	57.58%			
City inhabited by 20,000–100,000 people	33	26	55.93%			
City inhabited by more than 100,000 people	106	121	46.70%			
Education level					0.371	0.542
Secondary education	24	34	41.38%	1		
Higher education	176	184	48.89%			
Received diagnosis of DD						
Yes	186	202	47.94%	1	0.471	0.493
No	14	16	46.67%			

**Note.** ASD—autism spectrum disorders, DD—developmental disorders, *df*—degrees of freedom, and Χ^2^—chi-square test statistic result

**Table 3 brainsci-14-00388-t003:** Opinions on ASD screening (also online) and awareness of the problem among parents depending on parents’ age and IT proficiency.

Variables	H	*df*	*p*
Before screening, had you heard about ASD?			
Age	16.327	19	0.635
IT proficiency	14.801	5	0.011
Has it occurred to you that there may be a possibility that your children may have ASD?			
Age	12.970	19	0.840
IT proficiency	8.374	5	0.137
Should screening for DD be mandatory?			
Age	19.114	19	0.450
IT proficiency	3.277	5	0.657
Would you participate in screening for DD in your other children?			
Age	27.965	19	0.084
IT proficiency	7.550	5	0.183
Would you participate in online screening for DD in your other children?			
Age	16.013	19	0.656
IT proficiency	9.911	5	0.078
Do you prefer screening to be carried out stationary at your clinic?			
Age	9.599	19	0.962
IT proficiency	16.212	5	0.006

**Note**. ASD—autism spectrum disorders, DD—developmental disorders, *df*—degrees of freedom, and H—Kruskal–Wallis test statistic result.

**Table 4 brainsci-14-00388-t004:** Assessment of online ASD screening conducted in the “Spojrzeć w oczy” project by parents depending on place of residence and education level.

Variable	M	Me	SD	*df*	H	*p*
**The information available during screening was understandable and easily accessible**						
Place of residency					3.821	0.252
Village	4.773	5	0.516	3		
Town inhabited by less than 20,000 people	4.801	5	0.392			
City inhabited by 20,000–100,000 people	4.925	5	0.254			
City inhabited by more than 100,000 people	4.751	5	0.532			
Education level					1.317	0.517
Secondary education	4.811	5	0.381	1		
Higher education	4.781	5	0.504			
**I felt that I could refer any questions regarding my children’s development to the people responsible for screening**						
Place of residency					10.794	0.013
Village	4.350	5	0.896	3		
Town inhabited by less than 20,000 people	3.877	4	1.088			
City inhabited by 20,000–100,000 people	4.284	5	0.760			
City inhabited by more than 100,000 people	4.000	5	1.003			
Education level					1.710	0.452
Secondary education	3.900	5	1.165	1		
Higher education	4.144	5	0.927			
**I would receive appropriate help or advice in case of suspicion DD from people involved in online screening**						
Place of residency					21.323	<0.001
Village	4.433	5	0.778	3		
Town inhabited by less than 20,000 people	3.913	4	0.918			
City inhabited by 20,000–100,000 people	4.240	5	0.863			
City inhabited by more than 100,000 people	3.919	4	0.972			
Education level					6.043	0.048
Secondary education	4.264	5	0.935	1		
Higher education	4.051	4	0.925			
**Overall, I am satisfied with my participation in electronic ASD screening**						
Place of residency					3.450	0.278
Village	4.722	5	0.639	3		
Town inhabited by less than 20,000 people	4.769	5	0.415			
City inhabited by 20,000–100,000 people	4.705	5	0.544			
City inhabited by more than 100,000 people	4.655	5	0.607			
Education level					2.067	0.356
Secondary education	4.734	5	0.614	1		
Higher education	4.679	5	0.590			

**Note.** Answers were given on a Likert scale, where 1—I completely disagree with this sentence and 5—I completely agree with this sentence. ASD—autism spectrum disorders, DD—developmental disorders, *df*—degrees of freedom, H—Kruskal–Wallis test statistic result, M—mean, Me—median, and SD—standard deviation.

**Table 5 brainsci-14-00388-t005:** Assessment of online ASD screening conducted in the “Spojrzeć w oczy” project by parents depending on parents’ age and IT proficiency.

Variables	*r*	*t*	*p*
Age *&* The information available during screening was understandable and easily accessible	−0.093	−1.895	0.059
Age *&* I felt that I could refer any questions regarding my children’s development to the people responsible for screening	−0.104	−2.140	0.033
Age *&* I would receive appropriate help or advice in case of suspicion DD from people involved in online screening	0.033	0.677	0.499
Age *&* Overall, I am satisfied with my participation in electronic ASD screening	−0.030	−0.618	0.537
IT proficiency *&* The information available during screening was understandable and easily accessible	0.058	1.176	0.240
IT proficiency *&* I felt that I could refer any questions regarding my children’s development to the people responsible for screening	0.024	0.492	0.623
IT proficiency *&* I would receive appropriate help or advice in case of suspicion DD from people involved in online screening	−0.009	−0.185	0.853
IT proficiency *&* Overall, I am satisfied with my participation in electronic ASD screening	−0.026	−0.528	0.598

**Note**. ASD—autism spectrum disorders, DD—developmental disorders, *r*—Spearman’s rho statistic result, and *t*—the value of the *t* statistic testing the significance of the correlation coefficient.

**Table 6 brainsci-14-00388-t006:** Sociodemographic characteristics of doctors taking part in this study.

Characteristics (Total N = 95)	*n*	*%*
Sex		
Female	62	65.26%
Male	33	34.74%
Main place of practicing a profession		
Village	7	7.37%
Town inhabited by less than 20,000 people	9	9.47%
City inhabited by 20,000–100,000 people	22	23.16%
City inhabited by more than 100,000 people	57	60.00%
Percentage of pediatric patients among all doctor’s patients		
up to approx. 10% of total	22	23.15%
up to approx. 20% of total	27	28.42%
up to approx. 30% of total	19	20.00%
up to approx. 40% of total	6	6.32%
up to approx. 50% of total	10	10.53%
more than 50% of total	11	11.58%

**Table 7 brainsci-14-00388-t007:** The usage of screening methods, the choice of management in case of suspected DD and preferred screening method (online vs. stationary) depending on the sex, primary place of work of physicians and percentage of pediatric patients among all patients.

Variable	Total N = 95			Sex							Χ^2^/H*		*p*		Primary Place of Work													Χ^2^/H*	*p*
				Females N = 62			Males N = 33								Village N = 7			Town N = 9				Small City N = 22				Big City N = 57			
	n	%		n	%		n	%							n	%		n		%		n		%		n	%		
**Usage of screening methods for DD (e.g., ASD)**											1.246		0.265																
in all children	39	41.05%		28	45.16%		11	33.33%							2	28.57%		3		33.33%		7		31.82%		27	47.37%	2.388	0.496
in at-risk children or suspected of having DD	55	57.89%		33	53.23%		22	66.67%							5	71.43%		6		66.67%		15		68.18%		29	50.88%		
lack of use of screening tools	1	1.05%		1	1.61%		0	0.00%							0	0.00%		0		0.00%		0		0.00%		1	1.75%		
**Management of a child under 2 years of age in the event of subtle symptoms of developmental disorders**											0.897		0.343																
**referral for further diagnostics**	51	53.68%		32	51.61%		19	57.58%							2	28.57%		8		88.89%		13		59.09%		28	49.12%	1.162	0.762
referral for further diagnostics at the insistence of parents	8	8.42%		4	6.45%		4	12.12%							1	14.29%		0		0.00%		2		9.09%		5	8.77%		
further observation of the child’s development	36	37.89%		26	41.94%		10	30.30%							4	57.14%		1		11.11%		6		27.27%		24	42.11%		
**Management of a child above 2 years of age in the event of subtle symptoms of developmental disorders**											0.538		0.463																
referral for further diagnostics	90	94.74%		57	91.94%		33	100.00%							5	71.43%		9		100.00%		21		95.45%		55	96.49%	12.706	0.005
referral for further diagnostics at the insistence of parents	1	1.05%		1	1.61%		0	0.00%							1	14.29%		0		0.00%		0		0.00%		0	0.00%		
further observation of the child’s development	4	4.21%		4	6.45%		0	0.00%							1	14.29%		0		0.00%		1		4.55%		2	3.51%		
**Preferred screening method**																													
stationary	44	46.32%		30	48.39%		14	42.42%			0.304		0.581		3	42.86%		3		33.33%		6		27.27%		32	56.14%	6.001	0.112
on-line	51	53.68%		32	51.61%		19	57.58%							4	57.14%		6		66.67%		16		72.73%		25	43.86%		
**Variable**			**Percentage of Pediatric Patients Among All Patients**																									**Χ^2^/H***	** *p* **
			**<10%** **N = 22**						**<20%** **N = 27**			**<30%** **N = 19**					**<40%** **N = 6**				**<50%** **N = 10**				**>50%** **N = 11**				
			**n**			**%**			**n**	**%**		**n**		**%**			**n**		**%**		**n**		**%**		**n**		**%**		
**Usage of screening methods for DD (e.g., ASD)**																													
**in all children**			4			18.18%			18	66.67%		7		36.84%			0		0.00%		3		30.00%		7		63.64%	19.216	0.002
in at-risk children or suspected of having DD			18			81.82%			8	29.63%		12		63.16%			6		100.00%		7		70.00%		4		36.36%		
lack of use of screening tools			0			0.00%			1	3.70%		0		0.00%			0		0.00%		0		0.00%		0		0.00%		
**Management of a child under 2 years of age in the event of subtle symptoms of developmental disorders**																													
referral for further diagnostics			15			68.18%			12	44.44%		11		57.89%			3		50.00%		6		60.00%		4		36.36%	10.271	0.068
referral for further diagnostics at the insistence of parents			0			0.00%			1	3.70%		2		10.53%			0		0.00%		2		20.00%		3		27.27%		
further observation of the child’s development			7			31.82%			14	51.85%		6		31.58%			3		50.00%		2		20.00%		4		36.36%		
**Management of a child above 2 years of age in the event of subtle symptoms of developmental disorders**																													
referral for further diagnostics			20			90.91%			26	96.30%		19		100.00%			6		100.00%		9		90.00%		10		90.91%	8.590	0.127
referral for further diagnostics at the insistence of parents			0			0.00%				0.00%		0		0.00%			0		0.00%		1		10.00%		0		0.00%		
further observation of the child’s development			2			9.09%			1	3.70%		0		0.00%			0		0.00%		0		0.00%		1		9.09%		
**Preferred screening method**																													
stationary			13			59.09%			11	40.74%		5		26.32%			4		66.67%		6		60.00%		5		45.45%	6.524	0.259
on-line			9			40.91%			16	59.26%		14		73.68%			2		33.33%		4		40.00%		6		54.55%		

**Note.** ASD—autism spectrum disorders, DD—developmental disorders, H—Kruskal–Wallis test statistic result, and Χ^2^—chi-square test statistic result. * Kruskal–Wallis H test used only in preferred screening method analysis.

**Table 8 brainsci-14-00388-t008:** The usage of screening methods, the choice of management in case of suspected DD and preferred screening method (online vs. stationary) depending on the age, estimated number of children with DD under medical care and length of work experience of physicians.

Variables	H	*df*	*p*
Usage of screening methods for DD (in all children/only in children at risk or suspected of DD)			
age	22.435	17	0.169
self-estimated IT proficiency	5.496	6	0.482
estimated number of children with DD under medical care	22.133	18	0.226
length of work experience in primary care facilities	6.089	14	0.964
Management of a child under 2 years of age in the event of subtle symptoms of developmental disorders			
age	15.675	17	0.547
self-estimated IT proficiency	6.881	6	0.332
estimated number of children with DD under medical care	24.169	18	0.150
length of work experience in primary care facilities	24.553	14	0.039
Management of a child above 2 years of age in the event of subtle symptoms of developmental disorders			
age	18.120	17	0.381
self-estimated IT proficiency	19.757	6	0.003
estimated number of children with DD under medical care	13.250	18	0.777
length of work experience in primary care facilities	30.081	14	0.007
Preferred screening method			
age	27.876	17	0.046
self-estimated IT proficiency	5.277	6	0.501
estimated number of children with DD under medical care	24.435	18	0.141
length of work experience in primary care facilities	24.384	14	0.041

**Note.** DD—developmental disorders, *df*—degrees of freedom, and H—Kruskal–Wallis test statistic result.

**Table 9 brainsci-14-00388-t009:** Willingness to use DD screening diagnostic questionnaires during specific visits at primary care facilities.

How Willingly Would You Use Tools for Screening Developmental Disorders in the Following Situations?	M	Me	SD	H	*p*
during the vaccination qualifying visit at 18 months of age	3.512	4	1.094	46.069	<0.001
during the well-child visit in 2nd year of life	3.828	4	1.022		
when noticing symptoms of developmental disorders in child during the visit	4.415	5	0.736		
when a parent expresses concerns about their child’s development	4.362	5	0.735		
if the child has a sibling with ASD or another developmental disorder	4.207	4	0.787		

**Note.** Answers were given on a Likert scale, where 1—very reluctantly and 5—very willingly. ASD—autism spectrum disorders, H—Kruskal–Wallis test statistic result, M – mean, Me – median, SD – standard deviation,. Three respondents did not provide answers regarding this part.

**Table 10 brainsci-14-00388-t010:** Evaluation of the advantages and positives of online ASD screening by parents and physicians.

Variable	Parents N = 418			Doctors N = 95			Χ^2^ (*df* = 1)	*p*
	n	% of Parents	% of Total Answers	n	% of Doctors	% of Total Answers		
**Which of the positives of online ASD screening are the most important to you?**								
possibility of contact with healthcare professionals qualified in the field of developmental disorders	153	36.60%	13.08%	31	32.63%	13.19%	0.53	0.466
saving time on screening (possibility to perform screening at home)	285	68.18%	24.36%	61	64.21%	25.96%	0,56	0.456
no potential stigmatization of the child in the event of developmental disorders (person examining does not personally know the family/child)	118	28.23%	10.09%	30	31.58%	12.77%	0.42	0.515
possibility of performing the test at any convenient time	327	78.23%	27.95%	64	67.37%	27.23%	5.04	0.025
easier access to the screening (it is not necessary to look for people qualified to perform screening)	287	68.66%	24.53%	49	51.58%	20.85%	9.99	0.002
**Total**	1170	n/a	100%	235	n/a	100%		
**What are the greatest difficulties in conducting online screening in your opinion?**								
lack of direct (physical) contact with the examiner (e.g., doctor, nurse, psychologist)	195	46.65%	29.64%	53	55.79%	31.18%	2.59	0.108
inability to confirm the result of the screening test in clinical observation by a doctor/psychologist	242	57.89%	36.78%	50	52.63%	29.41%	0.87	0.350
the need to wait for an explanation of the obtained result	50	11.96%	7.60%	19	20.00%	11.18%	3.63	0.057
inability to quickly clarify doubts regarding the development of the child’s behavior	163	39.00%	24.77%	39	41.05%	22.94%	0.14	0.711
lack of trust in people responsible for online screening	8	1.91%	1.22%	9	9.47%	5.29%	11.55	<0.001
**Total**	658	n/a	100%	170	n/a	100%		

**Note.** ASD—autism spectrum disorders and Χ^2^—chi-square test statistic result.

## Data Availability

The data presented in this study are openly available in FigShare at https://doi.org/10.6084/m9.figshare.25465744.

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
