# Peer review of "Implementing an Early Detection Program for Autism Spectrum Disorders in the Polish Primary Healthcare Setting—Possible Obstacles and Experiences from Online ASD Screening"

_brainsci, 2024, doi:10.3390/brainsci14040388_

Round 1
Reviewer 1 Report
Comments and Suggestions for Authors
Thank you very much for giving me the opportunity to review this interesting manuscript titled “Implementing an early detection program for autism spectrum disorders in Polish primary healthcare setting – possible obstacles and experiences
from online ASD screening”.
With some improvements, the manuscript has the potential to be useful in the field. Overall the manuscript is appropriate for the scope of the Journal. The writing style is appropriate for a scientific manuscript. However, there are a few suggestions:
Abstract: The abstract should be revised according to the journal's instructions.
Introduction
- In line 43, the authors should add relevant references to their statement.
- In lines 47-50 the authors should also add text about the impact of ASD in family quality of life. Suggested references:
Papadopoulos A, Siafaka V, Tsapara A, et al. Measuring parental stress, illness perceptions, coping and quality of life in families of children newly diagnosed with autism spectrum disorder. BJPsych Open. 2023;9(3):e84. doi:10.1192/bjo.2023.55
Mello, C, Rivard, M, Terroux, A, Mercier, C. Quality of life in families of young children with autism spectrum disorder. Am J Intellect Dev Disabil 2019; 124(6): 535
- Lines 65-69 the authors should add references.
The authors provide a good overview and a clear statement of the research gap.
Methods: In this section, the authors provide details about the assessment of ASD electronic screening by the parents. However, the authors should revise this section and add more references that lead them to follow the specific methodology. The approach should also be more detailed to ensure the study can be replicated.
Also they describe appropriately the statistaical analysis that they followed. They used non-parametric test.
-in lines 153-157: This should be added to the limitations of the study.
Results
This section is well-presented, with details and a useful tables and 1 figure. The authors described adequately this section.
Discussion:
The lines 361-399 should transfer to the Introduction section or shorten. This section should start with a summary of the key findings before delving into the details and discuss the results. The authors discuss the study's findings appropriately.
Author Response
Thank you very much for giving me the opportunity to review this interesting manuscript titled “Implementing an early detection program for autism spectrum disorders in Polish primary healthcare setting – possible obstacles and experiences from online ASD screening”.
With some improvements, the manuscript has the potential to be useful in the field. Overall the manuscript is appropriate for the scope of the Journal. The writing style is appropriate for a scientific manuscript.
As authors, we are grateful for your time and commitment to reviewing our manuscript. We addressed your valuable comments later in the response.
However, there are a few suggestions:
Abstract: The abstract should be revised according to the journal's instructions.
We have slightly reformulated the abstract, adding more references to the study results. Due to the character limit in the abstract, we decided to somehow combine the results with conclusions.
Introduction: In line 43, the authors should add relevant references to their statement.
The appropriate literature reference has been added.
In lines 47-50 the authors should also add text about the impact of ASD in family quality of life. Suggested references:
Papadopoulos A, Siafaka V, Tsapara A, et al. Measuring parental stress, illness perceptions, coping and quality of life in families of children newly diagnosed with autism spectrum disorder. BJPsych Open. 2023;9(3):e84. doi:10.1192/bjo.2023.55
Mello, C, Rivard, M, Terroux, A, Mercier, C. Quality of life in families of young children with autism spectrum disorder. Am J Intellect Dev Disabil 2019; 124(6): 535
Thank you for recommending sources that will enrich the introduction of the manuscript - the data from the suggested studies have been briefly described by us in the introduction.
Lines 65-69 the authors should add references.
The appropriate literature reference has been added.
The authors provide a good overview and a clear statement of the research gap.
Methods: In this section, the authors provide details about the assessment of ASD electronic screening by the parents. However, the authors should revise this section and add more references that lead them to follow the specific methodology. The approach should also be more detailed to ensure the study can be replicated. Also they describe appropriately the statistical analysis that they followed. They used non-parametric test. -in lines 153-157: This should be added to the limitations of the study.
We supplemented the data from scientific articles that allowed us to create a research strategy; in addition, we added details regarding methodology, for example, verification of doctors or references to manuscripts describing the strategy for recruiting participants in a study on the psychometric properties of an ASD screening tool. We have also included information regarding the use of non-parametric tests or limitations in data analysis in the limitations.
Results: This section is well-presented, with details and a useful tables and 1 figure. The authors described adequately this section.
Discussion: The lines 361-399 should transfer to the Introduction section or shorten. This section should start with a summary of the key findings before delving into the details and discuss the results. The authors discuss the study's findings appropriately.
We fully agree with this comment - the fragment describing the method of organizing Polish well-child visits has been moved to the introduction, and a summary has been added.
Once again, we would like to thank you for your willingness to participate in the review process of our manuscript. We hope that our corrections, after taking into account your suggestions, have made the manuscript more scientifically valuable.
Reviewer 2 Report
Comments and Suggestions for Authors
The aim of the paper «Implementing an early detection program for autism spectrum disorders in Polish primary healthcare setting – possible obstacles and experiences from online ASD screening» was to assess the opinions of parents and physicians on the legitimacy and necessity of screening for autism spectrum disorders and identify the positive aspects and disadvantages of implementation screening programs, the possibility of using online screening for ASD.
The content of the рарer is relevant and important for medical science. The objectives of the study are clearly formulated. In the introduction, authors describe similar studies etiology of ASD, clinical symptoms and symptoms used for early diagnosis of the disease, questionnaires used and their capabilities. The methodology is described clearly, and the methods of statistical data analysis used are justified. The sample of study participants is sufficient - 418 families and 95 doctors. . Sociodemographic characteristics of parents taking part in the study were presented. But, as follows from Table 1, only 7.18% of children were diagnosed with DD (out of 30 - 21 children with ASD). The decision on mandatory screening for ASD was made primarily by parents of children without developmental disabilities. The results section provides a detailed analysis of the questionnaires for parents and doctors.
Strengths
1. The topic of the article is of practical importance, since it is aimed at improving the well-being of children through the introduction of screening for developmental disorders (DD) into medical practice.
2. Organization of the study.
3. Careful and detailed analysis of information provided by parents and doctors.
Weak Points
1. It would be important to add parents with children with ASD to the sample of parents.
2. Work experience (years) could be added to the questionnaire for doctors. The discussion noted that the doctors were predominantly resident-doctors. It is possible that including doctors with more experience and more conservative approaches would have yielded different results.
3. Discussion and Conclusion could be more concrete. The discussion is more like an Introduction; it is not enough for the received data. Also the discussion raises many more global issues than the specific research presented. Conclusion is more like Discussion.
General: In my opinion, this paper is not clearly suitable for the Brain science journal. The socio-medical focus of the research topic allows us to recommend it to International Research and Public Health journal ore another medical journal.
Author Response
The aim of the paper «Implementing an early detection program for autism spectrum disorders in Polish primary healthcare setting – possible obstacles and experiences from online ASD screening» was to assess the opinions of parents and physicians on the legitimacy and necessity of screening for autism spectrum disorders and identify the positive aspects and disadvantages of implementation screening programs, the possibility of using online screening for ASD.
The content of the рарer is relevant and important for medical science. The objectives of the study are clearly formulated. In the introduction, authors describe similar studies etiology of ASD, clinical symptoms and symptoms used for early diagnosis of the disease, questionnaires used and their capabilities. The methodology is described clearly, and the methods of statistical data analysis used are justified. The sample of study participants is sufficient - 418 families and 95 doctors. . Sociodemographic characteristics of parents taking part in the study were presented. But, as follows from Table 1, only 7.18% of children were diagnosed with DD (out of 30 - 21 children with ASD). The decision on mandatory screening for ASD was made primarily by parents of children without developmental disabilities. The results section provides a detailed analysis of the questionnaires for parents and doctors.
Thank you very much for your willingness to participate in the review process of our manuscript - we appreciate your devoted time and attention. Regarding the comment about small number of parents of children diagnosed with ASD - the validation process of the Polish CSBS-DP ITC questionnaire took place in the general population (i.e., depending on the prevalence data, from 0.2 to 1% of children may receive finally a diagnosis of ASD). In our validation sample, it included approximately 3% of children with ASD, and in the study assessing the online diagnostic process, over 7% of parents of children with ASD participated.
Strengths
- The topic of the article is of practical importance, since it is aimed at improving the well-being of children through the introduction of screening for developmental disorders (DD) into medical practice.
- Organization of the study.
- Careful and detailed analysis of information provided by parents and doctors.
We would like to thank you for your numerous positive comments regarding our manuscript.
Weak Points
- It would be important to add parents with children with ASD to the sample of parents.
Among a total of 418 parents, 21 parents of children with ASD participated in the study; their answers were also analyzed as, depending on the type of variables analyzed, as "parents of children without ASD vs parents of children with ASD" or "the entire group of parents including parents of children with ASD" (because ASD screening should be conducted in the general population).
- Work experience (years) could be added to the questionnaire for doctors. The discussion noted that the doctors were predominantly resident-doctors. It is possible that including doctors with more experience and more conservative approaches would have yielded different results.
The question regarding work experience is included in the questionnaire, although we only used work experience in a primary health care clinic - this is due to the fact that preventive tests on a group of children in Poland are only carried out in primary health care facilities. The analyzes conducted showed that more experienced doctors are more willing to refer children suspected of having developmental disorders for further evaluation.
- Discussion and Conclusion could be more concrete. The discussion is more like an Introduction; it is not enough for the received data. Also the discussion raises many more global issues than the specific research presented. Conclusion is more like Discussion.
Thank you for your suggestions - we have clarified the conclusion and shortened the discussion, also moving some of the discussion to the introduction. We decided to raise numerous issues related to the potential limitations of ASD globally to show that the features of health systems that cause these limitations can also be found in other countries, which means that our study may at least partially contribute to the discussion on screening in other countries than Poland.
General: In my opinion, this paper is not clearly suitable for the Brain science journal. The socio-medical focus of the research topic allows us to recommend it to International Research and Public Health journal ore another medical journal.
We submitted our paper to Brain Sciences because many of the sources we used to prepare our manuscripts on the topic of autism spectrum disorders were published in this scientific journal; for example, among many of the articles published in the special issue of Brain Sciences (https://www.mdpi.com/journal/brainsci/special_issues/Advance_Autism_Research) we can find those related to a socio-educational approach to ASD, especially due to the enormous impact of ASD on these issues. Moreover, due to the fact that this manuscript is to be part of a doctoral dissertation, we would like to publish the article in Brain Sciences, if possible, because this journal meets the internal criteria of the university where the doctoral thesis is being conducted.
Thank you again for your time spent reviewing our manuscript and your valuable comments. We hope that after the corrections, the article can be accepted for publication in Brain Sciences.
Round 2
Reviewer 1 Report
Comments and Suggestions for Authors
The authors made the revisions.